# The Combination of Radiation with PARP Inhibition Enhances Senescence and Sensitivity to the Senolytic, Navitoclax, in Triple Negative Breast Tumor Cells

**DOI:** 10.3390/biomedicines11113066

**Published:** 2023-11-16

**Authors:** Abrar Softah, Moureq R. Alotaibi, Ali R. Alhoshani, Tareq Saleh, Khalid Alhazzani, Mashal M. Almutairi, Raed AlRowis, Samiyah Alshehri, Norah A. Albekairy, Hisashi Harada, Rowan Boyd, Eesha Chakraborty, David A. Gewirtz, Homood M. As Sobeai

**Affiliations:** 1Department of Pharmacology and Toxicology, College of Pharmacy, King Saud University, Riyadh 11451, Saudi Arabia; 438203845@student.ksu.edu.sa (A.S.); mralotaibi@ksu.edu.sa (M.R.A.); ahoshani@ksu.edu.sa (A.R.A.); kalhazzani@ksu.edu.sa (K.A.); mmalmutairi@ksu.edu.sa (M.M.A.); saalshehri@ksu.edu.sa (S.A.); nalbekairi@ksu.edu.sa (N.A.A.); 2Department of Pharmacology and Public Health, Faculty of Medicine, The Hashemite University, Zarqa 13133, Jordan; tareq@hu.edu.jo; 3Department of Periodontics and Community Dentistry, College of Dentistry, King Saud University, Riyadh 11451, Saudi Arabia; dr.alrowais@gmail.com; 4Philips Institute for Oral Health Research, School of Dentistry, Virginia Commonwealth University, Richmond, VA 23298, USA; hharada@vcu.edu; 5Massey Cancer Center, Virginia Commonwealth University, Richmond, VA 23298, USA; david.gewirtz@vcuhealth.org; 6Department of Pharmacology and Toxicology, School of Medicine, Virginia Commonwealth University, Richmond, VA 23298, USA; boydk2@vcu.edu (R.B.); chakrabortye@vcu.edu (E.C.)

**Keywords:** senescence, PARP inhibitors, senolytics, breast cancer, radiotherapy, apoptosis

## Abstract

Despite significant advances in the treatment of triple-negative breast cancer, this disease continues to pose a clinical challenge, with many patients ultimately suffering from relapse. Tumor cells that recover after entering into a state of senescence after chemotherapy or radiation have been shown to develop a more aggressive phenotype, and to contribute to disease recurrence. By combining the PARP inhibitor (PARPi), talazoparib, with radiation, senescence was enhanced in 4T1 and MDA-MB-231 triple-negative breast cancer cell lines (based on SA-β-gal upregulation, increased expression of *CDKN1A* and the senescence-associated secretory phenotype (SASP) marker, *IL6*). Subsequent treatment of the radiation- and talazoparib-induced senescent 4T1 and MDA-MB231 cells with navitoclax (ABT-263) resulted in significant apoptotic cell death. In immunocompetent tumor-bearing mice, navitoclax exerted a modest growth inhibitory effect when used alone, but dramatically interfered with the recovery of 4T1-derived tumors induced into senescence with ionizing radiation and talazoparib. These findings support the potential utility of a senolytic strategy in combination with the radiotherapy/PARPi combination to mitigate the risk of disease recurrence in triple-negative breast cancer.

## 1. Introduction

Breast cancer is the most fdiagnosed malignancy among women worldwide [1,2,3,4]. One of the primary challenges for successful breast cancer therapy is disease relapse, which contributes significantly to higher mortality rates [5,6]. Therefore, developing a more effective therapeutic strategy designed to suppress disease recurrence is crucial for extending the survival of breast cancer patients. In part, resistance to chemotherapeutic agents or radiotherapy, both frequently utilized for the treatment of triple-negative breast cancer [5,7,8,9], can lead to the selection of aggressive tumor cell variants that could play a central role in cancer relapse [8,10,11,12]. Subsequently, elucidating mechanisms whereby cancer cells evade chemo- or radiotherapy is essential for the development of more effective means to combat cancer relapse.

Cellular senescence is a conserved cell stress response characterized by the suppression of cellular proliferation [8,12,13]. Promotion of senescence can result in response to a variety of stressors such as DNA damage, oncogene overexpression, mitochondrial dysfunction, and telomere shortening [8,14,15,16,17,18]. Of particular relevance to the current work, senescence is a fundamental response of tumor cells to various forms of anticancer therapeutics (both conventional and targeted), as well as exposure to radiation, which is often described as therapy-induced senescence (TIS) [19,20,21]. Furthermore, evidence of TIS has been identified in breast cancer patients exposed to neoadjuvant chemotherapy, suggesting its clinical relevance [22,23,24].

Despite their growth stagnation, senescent tumor cells remain metabolically active, survive further exposure to chemotherapy or radiation, and can sometimes evade growth arrest and resume proliferation [8,12]. Moreover, senescence has been shown to be directly linked to cancer relapse [25,26,27] and the development of more aggressive cancer phenotypes [8,19]. Therefore, the eradication of senescent tumor cells after therapy by forcing them into irreversible cell death (apoptosis) is proposed as an effective strategy to improve cancer therapy outcomes.

A heterogeneous group of pharmacological agents, collectively termed “senolytics” have the capability to selectively target and cull senescent cells [8,28,29,30]. Of those, navitoclax, a BCL-2/BCL_XL_/BCL_W_ inhibitor [31], was found to be one of the most effective agents to eliminate senescent cells, both in vitro and in vivo, in various aging-related disease models where senescence is implicated as a pathogenetic mechanism [32,33,34,35,36]. In cancer, navitoclax has also demonstrated a robust senolytic capability in tumor cell models triggered into TIS by doxorubicin [30,37], etoposide [38], cisplatin [39], gemcitabine [40,41], temozolomide [42], palbociclib [43,44], androgen deprivation [45], and radiation [46]. However, limited work has been published relating to the senolytic potential of navitoclax following TIS development by poly (ADP-ribose) polymerase (PARP) inhibitors.

PARP inhibitors (PARPi) have emerged as promising new therapies for triple-negative breast cancer (TNBC) associated with *BRCA1* and *BRCA2* mutations [47,48]. In addition to TNBC, other tumors, such as ovarian cancer, with homologous recombination deficiency, may also benefit from PARPi [49]. However, PARPi alone failed to show clinically significant improvement in the survival of patients with TNBC. Preclinical studies have shown a synergistic effect when PARPi are combined with radiation [50,51,52], but more importantly, as a potent inducer of TIS in tumor cells [53]. Interestingly, TIS that results from exposing tumor cells to the combination of radiation and PARPi has been described as unstable, and associated with a reversible form of growth arrest that is permissive for tumor cell proliferative recovery [52,54]. Thus, TIS by radiation and PARPi is hypothesized to contribute to cancer treatment failure and relapse [55].

In this work, two TNBC cell lines were utilized to examine the ability of the PARPi, talazoparib, to trigger TIS following radiation. Moreover, we sought to examine the ability of navitoclax to selectively eradicate senescent breast tumor cells in vitro and in tumor-bearing mice. Data from this work provides further evidence for the ability of senolytics to cull senescent tumor cells following exposure to cancer therapy.

## 2. Materials and Methods

### 2.1. Cell Lines and Drug Treatments

MDA-MB231 TNBC cancer cells, and 4T1 murine mammary carcinoma cells were cultured in Dulbecco’s Modified Eagle Medium (DMEM) with 10% fetal bovine serum (FBS) and 1% penicillin/streptomycin antibiotics from Gibco (Waltham, MA, USA). Talazoparib and navitoclax were purchased from Selleckchem (Houston, TX, USA). All drugs were suspended in dimethyl sulfoxide (DMSO) and administered in the dark at the desired quantities to achieve a final DMSO concentration in media (0.1%) or less. For all studies, unless indicated otherwise, TIS was established by pre-treating both cell lines with talazoparib (1 µM) for 3 h followed by 6 Gy radiation. Talazoparib was washed away from the medium 24 h after exposure to radiation (total talazoparib exposure time was 27 h). Navitoclax (2.5 µM) was applied to radiation- and PARPi-exposed cells on day 3 post-radiation.

### 2.2. Primers

Primers of the following were used in the study: *TP53* (tumor suppressor phosphoprotein) (IDs Hs01034249_m1, Mm01731290_g1), *CDKN1A* (cyclin-dependent kinase inhibitor) (IDs Hs00355782_m1, Mm00432448_m1), *IL6* (Interleukin-6) (IDs Hs00174131_m1, Mm00446190_m1), *CASP3* (Cysteine-aspartic acid protease) (IDs Hs00234387_m1, Mm01195085_m1), and *GAPDH* (glyceraldehyde-3-phosphate dehydrogenase) (IDs Hs02786624_g1, Mm99999915_g1). Primers were purchased from ThermoFisher Scientific (Waltham, MA, USA).

### 2.3. SA-β-Galactosidase Staining and C12-FDG Quantification to Evaluate Senescence

Senescence was confirmed in the treatment groups using the SA-β-galactosidase (SA-β-gal) histochemical staining and quantification of the SA-β-gal surrogate, C12-FDG, as described previously [56]. The SA-β-gal assay is a histochemical assay that involves the substrate, X-gal, in a staining buffer that is adjusted at pH 6.0 and is incubated overnight at 37 °C in a CO_2_-free incubator. Then, cells were washed with phosphate-buffered saline (PBS) and observed under bright field microscopy. Senescent cells were shown to be heavily stained in blue compared to non-senescent cells, with distinctive morphological features.

For C12-FDG staining and quantification, a fluorescence-based methodology was used to detect SA-β-gal activity via increasing the cellular pH to ~6 using a lysosomal inhibitory drug such as 100 nM of bafilomycin A1 for 1 h. Then, cells were incubated with 33 µM of C12-FDG, which is cleaved by β-galactosidase and emits fluorescence. Flow cytometry was used to detect and quantify the fluorescent signal. Senescent cells are expected to be highly C12-FDG positive while non-senescent cells are C12-FDG negative.

### 2.4. Annexin V/PI Assay to Measure Apoptosis

Apoptosis was evaluated in the experimental treated groups using the Annexin V/PI assay [57]. After the induction of apoptosis, cells were trypsinized and centrifuged. Then, medium supernatant was removed and the cell pellet was resuspended in 500 μL of 1× binding buffer containing 5 μL of annexin V-FITC and 5 μL of PI (BioLegend, San Diego, CA, USA). Samples were incubated at room temperature for 15 min in the dark. Finally, samples subjected to flow cytometry to measure Annexin V-FITC and PI signals. Four cell subpopulations were identified: non-apoptotic (Annexin V-negative/PI-negative), early apoptotic (annexin V-positive/PI-negative), late apoptotic (Annexin V-positive/PI-positive), and necrotic (annexin V-negative/PI-positive). We included both early and late apoptotic cell percentages in apoptosis calculations [52].

### 2.5. Reverse Transcription-Polymerase Chain Reaction (RT–PCR) to Measure Canonical Senescence-Related Gene Expression

The expression of senescence and apoptosis genes *TP53*, *CDKN1A*, *IL6*, and *CASP3* was investigated using RT-PCR [58]. Total RNA after treatment was extracted using the mRNeasy mini kit (Qiagen, Hilden, Germany) following the manufacturer’s protocol. Total RNA concentration was determined via measuring the absorbance at 260 nm in a spectrophotometer. RNA purity was obtained via calculating the ratio of the sample reading at 260 nm to 280 nm. Then, 2 μg of total RNA was converted to copy DNA (cDNA) using the high-capacity cDNA reverse transcription kit (Qiagen, Hilden, Germany). RNA was converted to cDNA in three thermal steps: 25 °C for 10 min, 37 °C for 120 min, then 85 °C for 5 min using a thermocycler. Primers of genes of interest were used to quantify gene expression. *GAPDH* was used to normalize gene expression data. Two-hundred ng of three technical replicates were run for each sample. The PCR process was run for 40 cycles of three thermal steps: 50 °C for 2 min, 9 °C for 10 min, then 95 °C for 15 s, followed by 60 °C for one minute.

### 2.6. Animal Tumor Model and Treatment

BALB/c female mice aged 5–6 weeks were maintained under pathogen-free conditions provided by the experimental animal facilities at the College of Pharmacy at KSU. First, 4T1 cells were grown for a few days in a 75 cm^2^ flask at 37 °C under a humidified environment with 5% CO_2_. Cells were exposed to talazoparib and radiation at in vitro settings to induce senescence, and then harvested according to the standard cell culture protocol. One million cells were suspended in PBS-Matrigel (1:1 dilution) and injected subcutaneously into the rear flank for each group (untreated and radiation plus talazoparib). Tumor growth was monitored daily. The treatment was initiated once the average tumor volume reached about 150–200 mm^3^. Then, the senolytic agent, navitoclax, was administered at 50 mg/kg via oral gavage every other day for a total of 3 doses, and tumor volume was measured with a Vernier caliper. Tumors were collected for further study.

### 2.7. Histopathological Analysis of Treatment Groups

Tumors were removed and fixed in 10% formalin for 1 week. After fixation, samples were dehydrated, cleared, infiltered, then embedded in paraffin wax. Blocks were sectioned at 6 µm, then dried in over 65 °C. Sections were stained with hematoxylin and eosin (H&E) and were photographed using a light microscope (Nikon, Tokyo, Japan) at 400×. Images were analyzed using ImageJ 1.54g (Bethesda, MA, USA) for degenerated cells count, 10 images/group.

### 2.8. TUNEL Assay of Tissue Samples in Response to Treatment

TUNEL Assay (TUNEL Assay Kit—BrdU-Red, Abcam, Cambridge, UK) was used as a convenient and sensitive method to detect DNA fragmentation as a marker of apoptosis [59]. This assay uses the terminal deoxynucleotidyl transferase (TdT) to catalyze the incorporation of deoxynucleotides at the free 3′-hydroxyl ends of fragmented DNA. The deoxynucleotides are then labelled in a variety of ways for the detection of the degree of DNA fragmentation. After fixing tissues with formaldehyde and washing, tissues were incubated with a proteinase K solution for 5 min at room temperature and refixed with formaldehyde. After an additional wash, tissues were incubated in DNA labelling solution for 60 min at 37 °C, followed by washing and incubation in the antibody solution for 30 min at room temperature. Then, the 7-AAD/RNase A solution was added followed by incubation for 30 min at room temperature, and analysis with fluorescence microscopy. A greater incorporation rate produces a brighter signal when the Br-dUTP sites are detected with an anti-BrdU monoclonal antibody directly labelled with a red fluorochrome. The BrdU-Red signal was analyzed at Ex/Em 488/576 nm, with an optional 7-AAD counterstain at Ex/Em 488/655 nm.

### 2.9. Statistical Analysis

All results were expressed as means ± standard errors. One-way ANOVA followed by Tukey’s post-hoc test was carried out to assess which treatment groups showed significant differences. The differences were considered significant when the *p*-value was less than 0.05. The statistical analyses were performed using GraphPad Prism 10 software Inc. (San Diego, CA, USA).

## 3. Results

### 3.1. PARP Inhibition Triggers Robust TIS in Combination with Radiation in MDA-MB-231 and 4T1 Breast Tumor Cells

Previous reports have indicated that the pre-treatment of irradiated tumor cells with PARPi robustly precipitates TIS in tumor cells with minimal induction of apoptosis [50,52,53]. Therefore, we sought to determine whether pre-treatment with talazoparib (1 µM) followed by radiation (6 Gy) would induce TIS in MDA-MB-231 and 4T1 breast cancer cells. Figure 1a,c demonstrate that the combination of talazoparib and radiation results in SA-β-gal upregulation and morphological changes consistent with senescence in both cell lines. Also, using the senescence surrogate, C12-FDG, to quantify the extent of SA-β-gal staining by flow cytometry, Figure 1b,d confirm that inclusion of the PARP inhibitor, talazoparib, along with radiation significantly increased the number of C12-FDG-positive cells compared to radiation alone. These data are commensurate with the previous literature, in that, combining PARPi and radiation results in robust TIS.

### 3.2. Navitoclax Enhances Apoptosis in Radiation and PARPi-Induced Senescence in MDA-MB-231 and 4T1 Breast Cancer Cells

Previous studies have established that senolysis by navitoclax is effective in sensitizing tumor cells to chemotherapy [46,60,61]. In this work, navitoclax (2.5 µM) alone promoted significant apoptosis in MDA-MB-231 cells (Figure 2a,b), but not in 4T1 cells (Figure 2c,d). Consistent with previous reports, and in line with our hypothesis, administration of navitoclax after the combination of PARPi and radiation synergistically enhanced apoptotic cell death in both cell lines (Figure 2b,d).

### 3.3. Navitoclax Modulates Expression of Senescence- and Apoptosis-Related Genes Induced by Radiation and PARPi

To further characterize the effect of navitoclax on breast tumor cells induced into TIS by the combination of radiation and PARPi, we performed qRT-PCR studies to investigate changes in expression in TIS- and apoptosis-related genes. To this end, we measured the expression of three genes (*TP53*, *CDKN1A*, and *IL6*) that are considered established indicators of senescence, and *CASP3* (caspase-3) as a marker for apoptosis. These studies revealed that radiation combined with talazoparib significantly upregulated the expression of all genes, except for the apoptosis-associated gene, *CASP3*, within 72 h post-treatment relative to the control (Figure 3). Furthermore, a 24 h treatment with navitoclax after radiation and talazoparib decreased the expression levels of the SASP component, *IL6* (Figure 3c). We also observed that the upregulation in *TP53* gene expression was further increased upon navitoclax treatment (Figure 3a), which indicates that cells endure severe genotoxicity and cellular stress post navitoclax therapy. Additionally, exposure to navitoclax following radiation and talazoparib led to a robust increase in *CASP3* expression, with minimal effect on *CDKN1A* levels (Figure 3b,d). Taken together, our data support the conclusion that the senolytic effect of navitoclax is coupled with decreased expression of senescence-related genes and overexpression of the apoptosis-related gene *CASP3*.

### 3.4. Navitoclax Suppresses Growth of Senescent 4T1 Cells in Tumor-Bearing Mice

The in vitro data presented above indicates that pretreatment of irradiated breast tumor cells with talazoparib results in pronounced TIS, and that subsequent treatment with navitoclax promotes apoptosis. To confirm these findings in an animal model, 4T1 murine breast cancer cells were pretreated with talazoparib (1 µM), followed by irradiation (6 Gy) and allowed to undergo TIS (~80%) for 72 h. Irradiated cells plus talazoparib and non-irradiated (and without talazoparib) cells were implanted into the flanks of BALB/C mice. Tumors cells were allowed to grow until they were palpable on day 4 post implantation. Navitoclax was then administered at 50 mg/kg p.o. for three doses every other day (i.e., days 5, 7, and 9). Figure 4a demonstrates that tumor cells exposed in vitro to radiation and talazoparib temporarily underwent growth inhibition, as did those exposed to navitoclax alone. However, the most potent effect produced by navitoclax was observed in mice challenged with 4T1 cells exposed to radiation and talazoparib, which led to a decline in tumor volume (tumor shrinkage), with no recovery throughout the monitoring period in this study.

Further studies were performed to interrogate the molecular outcomes associated with the studies presented in Figure 4a. A histopathological assessment of the harvested tumor samples using H&E staining showed that the untreated control tumors exhibited pleiomorphic shapes and open chromatin. In addition, the tumor samples that were exposed to radiation and talazoparib displayed degeneration of tumor cells leading to tissue loss and abundant hemorrhagic, necrotic areas (Figure 4b), accompanied by the destruction of cells by 40%, and were graded as IIa. The tumor samples treated with navitoclax alone exhibited mild destruction of tumor cells of about 35% and were graded as IIa (Figure 4b). Consistent with the tumor decline shown in Figure 4a, the tumor cells exposed to radiation and talazoparib followed by navitoclax showed extensive and pronounced degeneration, with tissues appearing congested and with necrotic debris and edema (Figure 4b), where the destruction of tumor cells was 59% and were graded as IIb (Table 1).

In line with the histopathological evaluation, the tumor samples were screened for apoptosis using the TUNEL assay. The data presented in Figure 4c show that tumor cells in the untreated group are well round and intact, indicating no evidence of apoptosis. On the other hand, tumor cells exposed to the triple-regimen of PARPi plus radiation followed by navitoclax clearly show DNA fragmentation and nuclear rupture, indicating that tumors originating from senescent 4T1 cells were more susceptible to apoptotic cell death induced by navitoclax. In contrast, both groups of tumors treated with radiation and talazoparib or navitoclax alone demonstrated minimal DNA fragmentation as well as reduced induction of apoptosis compared to those exposed to the triple-regimen. Taken together, the senolytic impact of navitoclax was found to be enhanced primarily in PARPi-irradiated tumors, while the effect of navitoclax in non-irradiated tumor cells was minimal.

## 4. Discussion

The treatment of TNBC has presented a continuing challenge. Although many advances have been achieved in terms of utilizing targeted therapy, TNBC is still relatively difficult to treat and more likely to recur in comparison with other types of breast cancers [9,62,63]. Therefore, we tested the “two-punch” approach in which TNBC cells are driven into a state of TIS to promote sensitivity to a senolytic agent. Previous reports from several laboratories have proven that using PARPi along with radiation would result in robust TIS [50,52,53]. Hence, we treated TNBC cells with talazoparib prior to radiation to trigger TIS, followed by a second hit with the senolytic agent, navitoclax, to eliminate senescent tumor cells.

In this work, we were able to establish a potent senolytic potential of navitoclax following TIS triggered by the exposure to radiation and the PARPi, talazoparib. One report has investigated a similar treatment approach utilizing a non-small-cell cancer model. Huart et al. exposed A549 tumor cells to X-rays (5 Gy as opposed to 6 Gy γ-radiation used in this work) and the PARPi, olaparib (as opposed to talazoparib in this work), and observed significant TIS marked by SA-β-gal upregulation and *CDKN1A* increased expression. In agreement with the results from our current study, Huart et al. showed that navitoclax was able to significantly induce apoptosis in radiation- and olaparib-treated cells, which was accompanied by a significant reduction in the gene expression levels of the SASP component, *IL6*, suggestive of a reduction in the burden of senescent tumor cells [61]. Despite confirming some data from this work, Huart et al. did not conduct in vivo studies to measure the senolytic potential of navitoclax following radiation and PARPi [61]. Of note, Huart et al. did not observe a cytotoxic effect of navitoclax in non-senescent tumor cells, which confirms its selectivity; however, we did detect some toxic effect of navitoclax in non-senescent MDA-MB-231 cells, likely related to the use of a relatively higher concentration (2.5 µM vs. 1 µM used by Huart et al.) and differences related to the studied cancer cell lines [61]. In other work, Fleury et al. demonstrated that the PARPi, olaparib, resulted in TIS in MDA-MB-231 TNBC cells in a concentration-dependent manner [54]. Interestingly, the use of olaparib in the absence of radiation produced a reversible form of TIS where senescent tumor cells resumed proliferation after cessation of drug exposure. More importantly, olaparib-induced senescence rendered breast tumor cells amenable to senolysis by navitoclax [54]. The senolytic effect of navitoclax was confirmed in vivo where its combination with olaparib resulted in reduced tumor volume at the end of a 12-day monitoring period [54]. While in vivo data from Fleury et al. were generated using human breast cancer xenografts in immunocompromised mice, our data confirm the senolytic potential of navitoclax in immunocompetent mice.

We also investigated the effect of navitoclax on the mRNA expression levels of senescence- and apoptosis-related genes. Our data confirmed the reduced gene expression levels of senescence-associated genes *TP53* (p53), *CDKN1A* (p21^Cip1^), and *IL6* (IL-6) and increased *CASP3* (caspase-3) levels. It is important to mention that the 4T1 cell line is p53-mutant [64,65], which could have influenced the expression of p53 and its downstream target *CDKN1A*. For instance, *CDKN1A* in the 4T1 cell line showed insignificant fold change among the treatment groups, with only a two-fold increase compared to the control group. However, mRNA expression of *CASP3* was significantly increased when navitoclax was added post to talazoparib and radiation. Several reports of different treatment models have shown that caspase activation and the Bax/Bcl-2 ratio increase is a common pathway of cytotoxicity in 4T1 cells [60,66,67]. Thus, our data suggest that the apoptotic cell death that was observed in the 4T1 cell line is highly attributed to the p53–caspase activation axis following Bcl-2 inhibition. However, the upregulation in *CASP3* needs to be further validated by measuring the cleaved capsase-3 protein expression. Moreover, we found that the SASP component, *IL6*, which plays an important role in inflammation, was greatly decreased following navitoclax exposure. Hence, we confirmed a reduction in inflammation as marked by the decrease in *IL6* gene expression along with the potent senolytic effect. Collectively, our data demonstrate that the employment of navitoclax following radiation and PARP inhibitors will sensitize TNBC.

In vivo results from this work showed that tumors originating from cells not exposed to radiation plus talazoparib or navitoclax alone demonstrated modest effects in terms of hemorrhage, necrosis, and apoptosis, which indicated limited cytotoxicity (Figure 4b,c). In contrast, tumors originating from cells exposed to radiation plus talazoparib followed by navitoclax showed strongly degenerative, necrotic, and apoptotic effects (Figure 4b,c). Saleh et al. [30] have previously reported that navitoclax post to a high dose of radiation (10 Gy) in vivo showed similar outcomes; however, we show here that the inclusion of PARP inhibitors would allow for the use of a lower dose of radiation and promote more robust TIS and greater susceptibility to navitoclax, which would minimize the side effects expected from the use of high-dose radiation. In addition, the significant reduction in tumor growth after the treatment with navitoclax indirectly suggests that the drug mediates cell cycle arrest or prolonged cell cycle re-entry of these senescent cells relative to untreated senescent cells. However, future work is warranted to examine the direct impact of navitoclax on the cell cycle utilizing flow cytometry-based functional assays and/or measuring the expression of cell cycle biomarkers using western or immunofluorescence techniques. Nevertheless, a major limitation from this work is that the development of TIS in the 4T1 cell lines was performed in vitro using radiation and talazoparib treatment prior to implantation of the tumor cells. Despite the consistency of navitoclax’s senolytic effect, this methodological approach could have influenced the tumor microenvironment changes that could accompany the development of TIS by directed drug or radiation delivery and the promotion of TIS in mice.

The rationale that supports the use of senolytics as complementary treatments for cancer is primarily based on the fact that the persistence and accumulation of therapy-induced senescent cells has been strongly linked to cancer relapse and several untoward outcomes of chemo- or radiotherapies [27]. For example, TIS has been shown to contribute to the development of doxorubicin-induced cardiotoxicity and cisplatin-induced peripheral neuropathy in mouse models [68,69]. Unfortunately, the ultimate fate of senescent tumor cells in vivo is still poorly understood. At least in vitro, senescent growth arrest in tumor cells can be overcome and might allow for the re-emergence of proliferating tumor cells. Recent evidence has suggested that a subpopulation of senescent tumor cells can evade growth arrest and resume proliferation [70,71]. Subsequently, the culling of senescent tumor cells perhaps could reduce the potential for cancer relapse and mitigate the adverse outcomes of chemoradiation. Unfortunately, despite being a powerful senolytic, navitoclax is associated with treatment-limiting thrombocytopenia. Interestingly, Estepa-Fernández et al. employed a novel technology of navitoclax-loaded nanoparticles that were coated with a hexa-oligo-saccharide to destroy senescent tumor cells with a high degree of selective targeting and reduced adverse effects [72]. Accordingly, pharmaceutical improvements to some of the most tested senolytics might accelerate their investigation in the clinic for cancer treatment. Finally, the use of navitoclax and other senolytics should be considered after careful individualization, and potentially only for patients with a high burden of TIS at the end of their chemo- or radiotherapy treatment, especially considering that the extent of TIS in vivo is still undetermined and can largely vary.

## 5. Conclusions

In conclusion, we demonstrated that the treatment regimen of a PARP inhibitor, talazoparib, with an effective dose of radiation followed by the senolytic agent, navitoclax, was efficient in the elimination of senescent TNBC cells by shifting them to apoptosis, which was validated in an immunocompetent tumor-bearing mouse model. Our study provides support for the potential use of navitoclax with the radiotherapy/PARPi combination as a novel therapeutic strategy to improve patient outcomes and mitigate the risk of breast cancer recurrence. Our work paves the road for future experimental work investigating in depth the impact of the treatment region presented in the study on the cell senescence signaling pathway and SASP. In addition, preclinical safety studies of the examined treatment regimen are warranted.

## Figures and Tables

**Figure 1 biomedicines-11-03066-f001:**
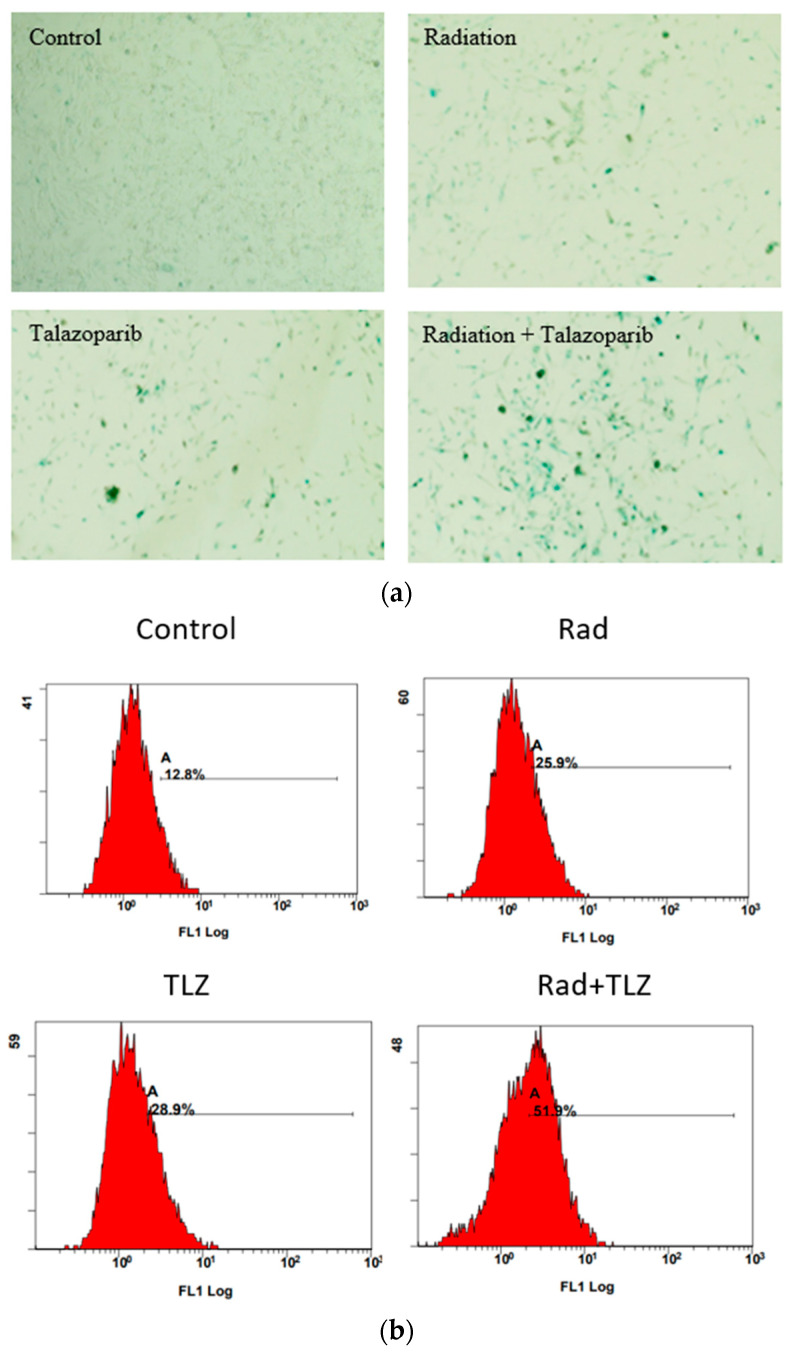
Pre-treatment with PARP inhibitors induces robust senescence in irradiated MDA-MB-231 and 4T1 breast cancer cells. β-galactosidase upregulation was monitored in MDA-MB-231 and 4T1 breast cancer cells to evaluate senescence markers post-radiation (6 Gy). (**a**,**d**) Images show high expression of β-galactosidase in both cell lines when exposed to radiation post-PARP inhibition. (**b**,**e**) illustrate representative flow cytometry charts of senescent cell percentage using C_12_FDG staining. A shows the gated C_12_FDG-postive cells that were considered senescent cells. (**c**,**f**) Evaluation of senescence in both cell lines using flow cytometry with C_12_FDG. Values are expressed as mean ± SEM. * *p* < 0.05, ** *p* < 0.01, *** *p* < 0.001, and **** *p* < 0.0001 indicate statistical significance between treatment groups (one-way ANOVA followed by Tukey’s post-hoc test; *n* (number of independent experiments) = 3).

**Figure 2 biomedicines-11-03066-f002:**
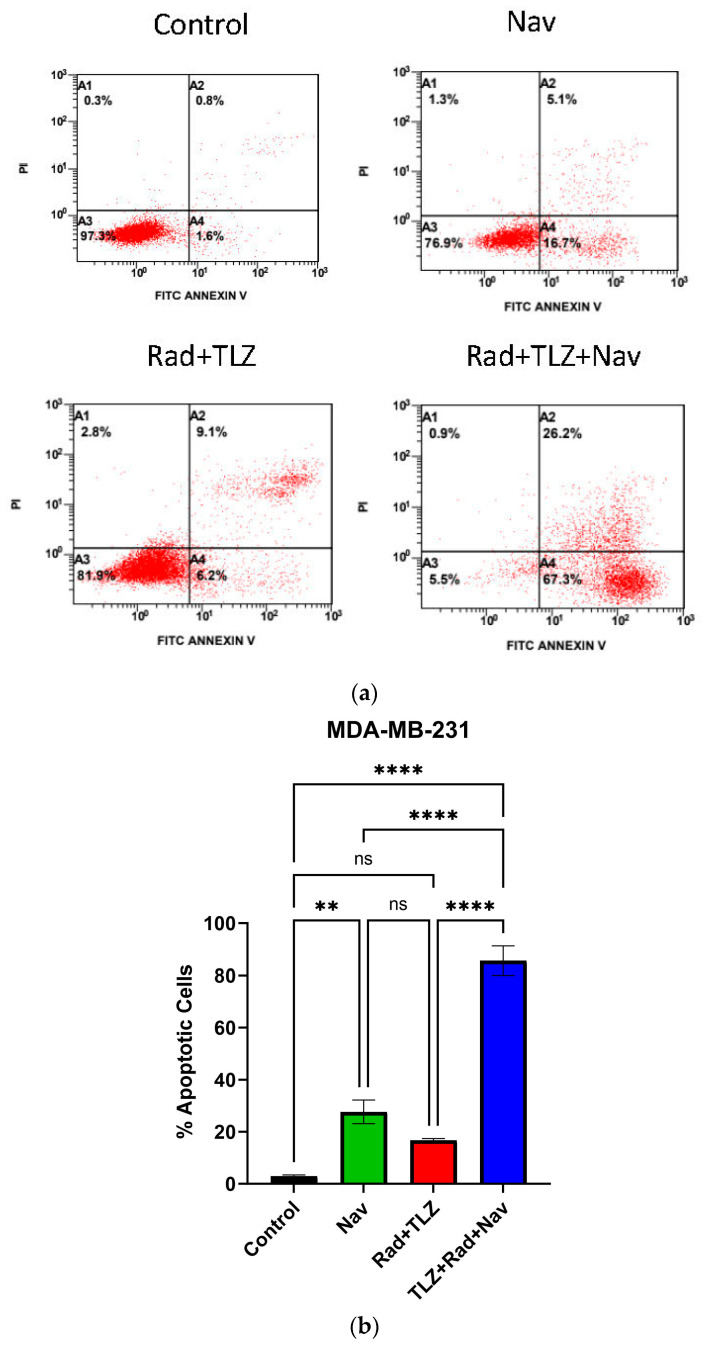
Addition of navitoclax to radiation and PARP inhibitor promotes apoptosis in MDA-MB-231 and 4T1 breast cancer cells. Both cell lines were treated with navitoclax, radiation and talazoparib, and radiation and talazoparib followed by navitoclax. Cell apoptosis was examined based on Annexin V/PI staining using flow cytometry. Four subpopulations have been detected, A1: dead cells (annexin V−/PI+), A2: late apoptotic cells (annexin V+/PI+), A3: viable cells (annexin V−/PI−), A4: early apoptotic cells (annexin V+/PI−). Percentages of apoptotic cell subpopulations were computed, including both early and late apoptotic cells. Panels (**a**,**b**) show apoptosis profile in the MDA-MB-231 cell line. Panels (**c**,**d**) represent apoptosis profile in the 4T1 cell line. Values are expressed as mean ± SEM. * *p* < 0.05, ** *p* < 0.01, and **** *p* < 0.0001 indicate statistical significance between treatment groups (one-way ANOVA followed by Tukey’s post-hoc test; *n* (number of independent experiments) = 3).

**Figure 3 biomedicines-11-03066-f003:**
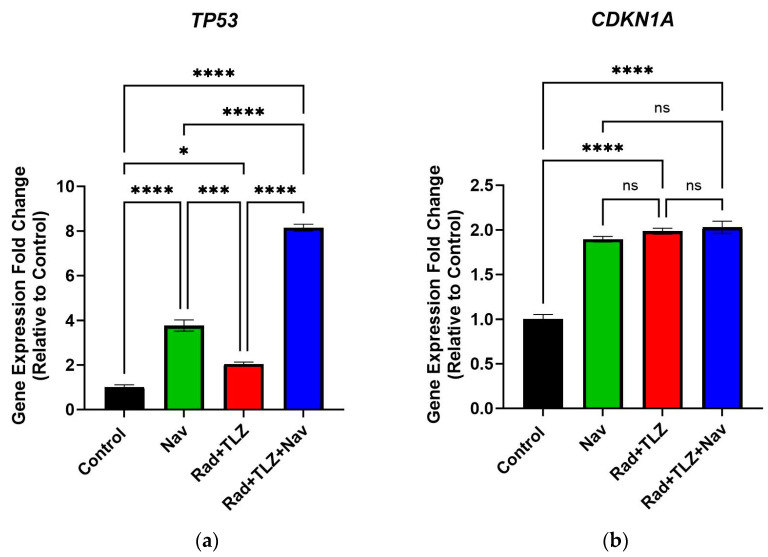
Determination of gene expression of apoptosis and senescence genes in 4T1 breast cancer cells. Isolated RNAs of four treatment groups were assessed for *TP53*, *CDKN1A*, *IL6*, and *CASP3* expression for 4T1 cell line. (**a**) Shows data for the expression of *TP53*, (**b**) shows data for the expression of *CDKN1A*, (**c**) shows data for the expression of *IL6*, and (**d**) shows data for the expression of *CASP3*. Values are expressed as mean ± SEM. * *p* < 0.05, ** *p* < 0.01, *** *p* < 0.001, and **** *p* < 0.0001 indicate statistical significance between treatment groups (one-way ANOVA followed by Tukey’s post-hoc test; *n* (number of independent experiments) = 3).

**Figure 4 biomedicines-11-03066-f004:**
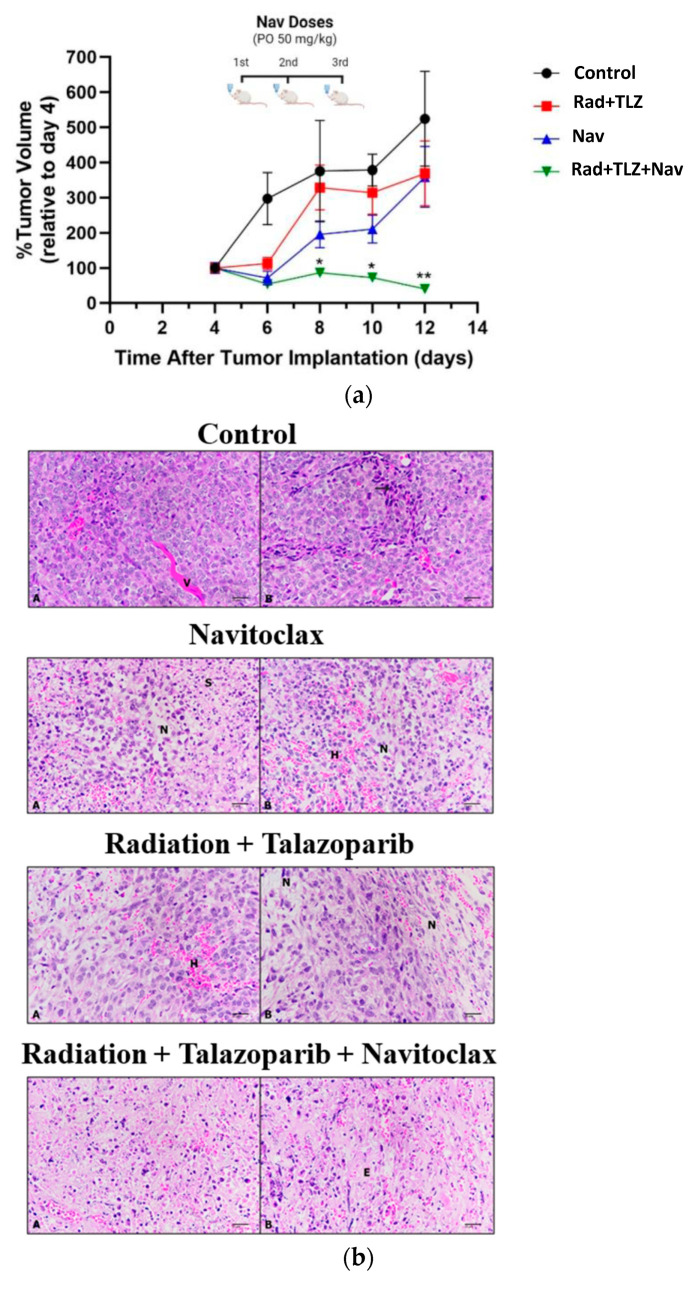
Assessment of senolytic effect of navitoclax in 4T1 breast cancer cells following radiation and talazoparib in mice. 4T1 cells were exposed to 6 Gy following 1 µM talazoparib at in vitro settings to induce senescence; then cells were implanted into Balb/c mice. The mice were then treated with navitoclax 50 mg/kg (p.o.) every other day for 7 days (four doses). (**a**) Growth curves for untreated tumor cells (left panel) and irradiated PARPi cells (right panel) with or without navitoclax. (**b**) Histopathological screening of tumor samples using H&E staining (H&E-400×). Photomicrographs of untreated tumor, radiation plus talazoparib, navitoclax alone, and radiation plus talazoparib followed by navitoclax are displayed. (**A**) Loose tumor cells, (V) blood vessels, (**B**) inflammatory cells inside the tumor, (H) hemorrhage, (N) necrotic areas, (E) edema, and (S) stroma as indicated in the images. (**c**) TUNEL images of the in vivo tissue samples. * *p* < 0.05 and ** *p* < 0.01, indicate statistical significance between treatment groups (one-way ANOVA followed by Tukey’s post-hoc test; *n* (number of mice in each experimental group) = 6).

**Table 1 biomedicines-11-03066-t001:** Viable cells count of tumor, destruction %, and Evan’s grade in the experimental groups. Data are represented as mean ± SEM, *p* < 0.05 considered significant.

Groups	Viable Cells Count	Destruction %	Evan’s Grade
Control	3997 ± 214	2	I
Radiation + talazoparib	2418 ± 205 *^a^	40	IIa
Navitoclax	2624 ± 191 *^a,^*^b^	35	IIa
Radiation + talazoparib + navitoclax	1664 ± 66 *^a,^*^b^	59	IIb

*^a^ significant relative to control. *^b^ significant relative to radiation and talazoparib.

## Data Availability

Data sets generated in this study are available on reasonable request to the corresponding author.

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
