# Peer review of "The Combination of Radiation with PARP Inhibition Enhances Senescence and Sensitivity to the Senolytic, Navitoclax, in Triple Negative Breast Tumor Cells"

_biomedicines, 2023, doi:10.3390/biomedicines11113066_

Round 1

Reviewer 1 Report

Comments and Suggestions for Authors

In a study by Softah et al., the Authors investigate the senolytic effect of navitoclax in breast cancer cells. Although the concept of this study is interesting and clinically relevant, technical quality of the results and their presentation are very poor. As such, this manuscript requires a thorough revision to be suitable for publication, and the experimental part should be extended to support the mechanistic view of these results.

Specific comments:

1. Title is unclear ("senolytic effect .... induced into senescence"?)

2. Line 57 - please correct that navitoclax is an inhibitor of BCL-2, BCL-XL and BCL-w. 

3. Introduction is a combination of several pieces of information with a poor connection between them.

4. Paragraph 2.2. - antibodies are not primers. In addition, primer sequences or catalog numbers should be included.

5. Fig. 1 - scale bars are missing. Why was statistical significance calculated only between radiation group and combination group?

6. Fig. 2 - similar concern on the statistical comparison as above.

7. Fig. 3 - selection of genes for analysis seems to be random and does not follow any justified reasons. In addition, assessing the SASP program would benefit from assessment of secreted IL-6.

8. Discussion is mostly the repetition of the results. Detailed reference to earlier papers would improve the quality and significance of this study.

Comments on the Quality of English Language

minor revision

Reviewer 2 Report

Comments and Suggestions for Authors

The study presents an integrated therapeutic approach aimed at mitigating the risk of breast cancer recurrence. It concentrates on eliminating senescent tumor cells through the combined use of PARP inhibitors (namely Talazoparib), ionizing radiation, and the senolytic agent Navitoclax. The research provides promising results in both cell culture and animal models. However, there are specific areas that could benefit from further elucidation:

1. It would be valuable to incorporate flow cytometry plots for C12-FDG staining, particularly since this method was used for quantification rather than immunofluorescence for beta-Galactosidase.

2. The study would be strengthened by the inclusion of additional senescence markers, such as p16, to bolster the findings. This would offer a more robust evidence base than relying solely on a single SASP gene.

3. It is advisable for the authors to measure the levels of cleaved caspase 3 using Western blot analysis, as this protein is better indicator of apoptosis.

4. For further clarity, the study should specify the criteria used to categorize cells as apoptotic. For example, are cells considered apoptotic based on being both Annexin V and PI positive, or solely Annexin V positive?

Reviewer 3 Report

Comments and Suggestions for Authors

The article titled "Senolytic Effect of Navitoclax in Breast Cancer Cells Induced into Senescence by Radiation and PARP Inhibition" by Abrar Softah et al. provides valuable insights, but there are several issues that need to be addressed:

1. The author should rewrite the abstract to clearly state the background, aim of the study, materials and methods, results, and conclusion without using overly complex sentences. It is also important to include the study objectives in the abstract and define any abbreviations used.

2. It is noted that the manuscript with the abstract has been published in the 47th FEBS-2023 congress (https://www.2023.febscongress.org/abstract_preview.aspx?idAbstractEnc=4424170096091091100099424170).

3. The authors may consider performing a colony formation assay using with or without Navitoclax in radiation and PARP inhibitor-induced senescence in TNBC cells to determine the role of Navitoclax.

4. The authors may consider performing cell cycle marker assays using with or without Navitoclax in radiation and PARP inhibitor-induced senescence in TNBC cells. This can be achieved by using western blot or immunofluorescence methods to detect cell cycle marker proteins.

5. Additionally, the authors may consider performing cell viability tests using MTT assays in TNBC cells with or without Navitoclax.

6. The discussion should include an interpretation of the results in the context of previous studies and hypotheses. The implications of the findings should be discussed in a broad context, and the limitations of the study should be highlighted. Future research directions should also be mentioned.

Comments on the Quality of English Language

Moderate editing of English language required

Round 2

Reviewer 1 Report

Comments and Suggestions for Authors

The Authors have substantially improved the manuscript. However, I have still concerns of the title. In my opinion, the title is not correct, looking at least from the language point of view. Especially the part "induced into senescence" is unclear.

Reviewer 3 Report

Comments and Suggestions for Authors

The authors did not conduct any of the suggested experiments, such as the colony formation assay, cell cycle marker assay, or MTT assay, to validate or support the hypothesis.

Comments on the Quality of English Language

Moderate editing of English language required
